# Feasibility of a Sensor-Based Technological Platform in Assessing Gait and Sleep of In-Hospital Stroke and Incomplete Spinal Cord Injury (iSCI) Patients

**DOI:** 10.3390/s20102748

**Published:** 2020-05-12

**Authors:** Maartje M. S. Hendriks, Marije Vos-van der Hulst, Noel L. W. Keijsers

**Affiliations:** 1Department of Research, Sint Maartenskliniek, Hengstdal 3, 6574 NA Ubbergen (near Nijmegen), The Netherlands; n.keijsers@maartenskliniek.nl; 2Department of Rehabilitation, Donders Institute for Brain, Cognition and Behaviour, Radboud University Medical Center, Heyendaalseweg 135, 6525 AJ Nijmegen, The Netherlands; 3Department of Rehabilitation, Sint Maartenskliniek, Hengstdal 3, 6574 Ubbergen (near Nijmegen), The Netherlands; m.vos-vandehulst@maartenskliniek.nl

**Keywords:** sensor technology, technological platform, feasibility, clinical implementation, stroke, spinal cord injury, rehabilitation, gait, sleep

## Abstract

Recovery of the walking function is one of the most common rehabilitation goals of neurological patients. Sufficient and adequate sleep is a prerequisite for recovery or training. To objectively monitor patients’ progress, a combination of different sensors measuring continuously over time is needed. A sensor-based technological platform offers possibilities to monitor gait and sleep. Implementation in clinical practice is of utmost relevance and has scarcely been studied. Therefore, this study examined the feasibility of a sensor-based technological platform within the clinical setting. Participants (12 incomplete spinal cord injury (iSCI), 13 stroke) were asked to wear inertial measurement units (IMUs) around the ankles during daytime and the bed sensor was placed under their mattress for one week. Feasibility was established based on missing data, error cause, and user experience. Percentage of missing measurement days and nights was 14% and 4%, respectively. Main cause of lost measurement days was related to missing IMU sensor data. Participants were not impeded, did not experience any discomfort, and found the sensors easy to use. The sensor-based technological platform is feasible to use within the clinical rehabilitation setting for continuously monitoring gait and sleep of iSCI and stroke patients.

## 1. Introduction

Neurological injury (e.g., stroke or incomplete spinal cord injury (iSCI)) often leads to motor impairment, which affects daily activity and mobility (walking) [1,2,3]. Recovery of walking function is considered of utmost relevance and is one of the most common rehabilitation goals of neurological patients [4,5]. Moreover, early ambulation activity during rehabilitation has shown to have many beneficial effects on physical health, such as increased muscle strength and improved endurance [6,7]. Furthermore, several studies demonstrated that physical activity during the early rehabilitation period positively affects an individual’s ultimate functional capacity [8,9]. However, patients with stroke and iSCI have a relatively low level of physical activity and have difficulty in initiating activity [10,11]. Therefore, in-hospital rehabilitation aims to increase patients’ physical activity, both during rehabilitation therapy, as well as during the remainder of the day [11]. In addition to physical activity during the day, sufficient and adequate sleep also promotes rehabilitation by gaining physical and psychological energy [12,13].

Several methods have been used to assess patients’ physical activity and sleep. Most of the time these methods rely on self-report, making them subjective and less accurate [14,15,16]. A more objective way for assessing patients’ physical activity is by using inertial measurement units (IMUs), which can measure posture, movement, and spatiotemporal gait parameters [2,17,18,19,20,21]. IMUs are user-friendly with limited burden for the patient and therefore seen as a convenient way to measure physical activity [22]. For sleep assessment, electroencephalography (EEG) and polysomnography (PSG) are the gold standard. However, these measures are experienced impractical in a clinical setting [13,14,23]. An alternative to determine sleep quality objectively is a contactless bed sensor under the mattress based on ballistocardiography (BCG) [23]. BCG technology consists of a highly sensitive pressure sensor which can measure heart rate (HR), respiratory rate (RR), and body movements (turns, tosses, and bed exits) which are parameters used in sleep monitoring [24,25]. Therefore, a combination of a bed sensor and IMUs seems feasible to objectively monitor patients 24 h a day.

To date, studies using wearable sensors are mostly conducted in controlled environments and laboratories under close observation of for example a researcher or therapist. However, results measured in such controlled environments do not necessarily represent what patients do outside those standardized settings [26]. To objectively measure a patient’s recovery and participation level in the rehabilitation process, a combination of different sensors measuring continuously over a longer period of time could be useful [2,12]. Many sensors have been developed and tested in research settings, however, there is a lack of real clinical implementation of those sensors [27,28,29]. To implement multiple sensor technologies in clinical routine, a system is needed that can record, store and analyze data, and subsequently provide feedback to medical personnel in a way that is most appropriate for the clinical setting [30]. 

A technological platform offers the possibility to incorporate those features. By developing such a platform, clinicians can eventually provide an optimized and personalized treatment plan for the patient [18,22]. For developing such a platform, modern machine learning algorithms based on real-time patient-specific movement data are needed and can be gathered by continuously tracking gait and sleep activity [31]. Although the development is a prerequisite, the implementation of such a platform in clinical practice is a critical part for success and effective implementation is not a straightforward process. Various factors are involved including system performance, usability, and integration with existing work practices [32,33,34]. Therefore, the aim of this study was to develop and test the feasibility of a technological sensor-based platform in clinical rehabilitation practice.

The overall purpose of this project was to determine how a technological sensor-based platform can be integrated within the existing clinical treatment regimen. Herewith, the first step was to test the feasibility of using sensor technology within the clinical setting based on missing data, error cause, and user experience. This study was an iterative process with human as well as technological factors involved. It was hypothesized that our sensor-based technological platform is feasible to use in a clinical setting to continuously measure gait and sleep. This paves the way for further developments towards the actual implementation of the sensor-based technological platform on the rehabilitation ward. This article mainly focused on the feasibility of the platform in which the bed sensor was not completely integrated. The feasibility of the bed sensor will be discussed in another article.

## 2. Materials and Methods

### 2.1. Study Design

This observational study was carried out at the Sint Maartenskliniek (Ubbergen, The Netherlands). The study was conducted as part of the “Smarten the Clinic” project, which was approved by the Medical Ethical Research Committee of Arnhem-Nijmegen (4222-2018). The interaction between participant and researcher was kept to a minimum to prevent interference with clinical practice. 

### 2.2. Study Population

Patients with an iSCI with an American Spinal Injury Association (ASIA) scale [35] of C/D/E and first-ever stroke patients with a Functional Ambulation Category (FAC) score [36] of ≥2 participated in the study. Additional inclusion criteria were; minimum age of 16 years, not wheelchair-bounded, participation in ambulation therapy, no other comorbidities affecting patients’ ambulatory function, and no use of an anti-decubitus air mattress because of interference with the measurement. Participants who were unable to grant permission to participate in the study due to language issues or cognitive impairment were excluded. Inclusion of participants stopped if each group consisted of twelve participants who participated for an entire measurement week. Written informed consent was obtained prior to participation.

### 2.3. Technological Platform Architecture

#### 2.3.1. Hardware

The hardware design of the technological platform consisted of the sensors (two IMUs and a bed sensor) and a docking station (Figure 1). The docking station comprised of two docks (Shimmer Dock; Shimmer, Dublin, Ireland) which were placed on a small computer (Next Unit of Computing (NUC)). The docks were connected to the NUC via a USB cable. Gait data were obtained by using Shimmer^®^3 IMUs (Shimmer, Dublin, Ireland; 51 mm × 34 mm × 14 mm, 23.6 g). The IMUs were placed on the docking station for charging and data transfer. Tri-axial accelerometer, gyroscope, and magnetometer data were collected with a sample frequency of 102.4 Hz. Bed sensor data were obtained using the Emfit QS sleep tracker (Emfit ltd., Helsinki, Finland), a wireless and contactless pressure sensor which was placed under the patient’s mattress. 

#### 2.3.2. Software

Software design of the technological platform included four steps: data upload, pre-processing, processing, and storage of the IMU data (Figure 1). Data upload by the docking station was an event-driven architecture with on-demand computation and storage in the Azure cloud (Microsoft Azure Storage Explorer). Full integration of the bed sensor into the technological platform was not established during this study. Sleep data were stored automatically on the Emfit server accessible at qs.emfit.com.

##### Data Upload

Data processing was initiated by the placement of the IMUs on the docking station. This triggered the python data copy program installed on the NUC to combine the raw data of the left and right legs. The combined data were uploaded to the cloud for pre-processing. When uploading to the Cloud was completed, the pre-processing was initiated automatically.

##### Data Pre-processing

Pre-processing contained four aspects; Storage Blobs, which were used to save the raw data in a zip-format for easier processing; an Event Grid, which called the Azure function when the upload was completed; an Azure Function, which created a processing job in the Azure Batch Service containing a link to the zip-file to be processed; and the Batch Service, which deployed a (new) Virtual Machine, prepared the environment on the machine and retrieved the zip-file to be processed. After pre-processing was completed, the Azure Batch uploaded the results and the log files and deleted the Virtual Machine. 

##### Data Processing

Data were processed using Java and R. Firstly, the raw data in the zip-file were converted by a Java script to comma-separated values (CSV) data. Subsequently, those CSV data were processed by KIDUKU’s algorithms (Fujitsu Laboratories Ltd. and Fujitsu Limited) in R to extract and calculate key features of gait (gait velocity, step count, stride length). This fully processed data is referred to as gait data.

##### Data Storage

Sensor data in the zip-file, CSV data, and gait data were stored in a Microsoft^®^ Azure Platform. During the study, there was no integration of the gait analysis results into a dashboard interface, but this would be a logical next step.

### 2.4. Experimental Protocol

The aim was to monitor the gait of the participants for seven consecutive days. During the weekend, patients could leave the clinic for one or two days resulting in a measurement week consisting of a minimum of five to a maximum of 7 measurement days. For practical reasons it might happen that a measurement week contained an eighth day. Participants followed their normal rehabilitation program during the measurement week. The bed sensor was placed under the mattress of the participant at the level of the thorax. The docking station was located near the participant’s bed. Participants were instructed to wear the IMUs on both ankles during the daytime. In the morning, they had to place the IMUs on the lateral side of each ankle just above the malleolus (Figure 2). In the evening, just before participants went to sleep, the IMUs had to be placed back on the docking station. During swimming, showering, or bathing, the IMUs had to be taken off because they are not water-resistant. If necessary, the nursing staff assisted in handling the IMUs in the morning and/or evening. At the end of the measurement week, participants filled out a questionnaire about their experience with the sensors. Interaction between participant and researcher occurred only at the start and end of the measurement week, during two prespecified walking sessions, and if uncertainties or problems were reported by nursing staff and/or patients.

### 2.5. Analysis and Outcome Measures

The primary outcome to evaluate the feasibility of the technological platform was the percentage of missing measurement data (days and nights). A day or night was considered as missing when data were expected but weren’t available. In addition, the clinical applicability (reasons for missing measurement days and nights) in terms of personal responsibility and technological shortcomings was determined. Causes of missing and lost measurement data can roughly be divided into three categories; human interaction errors, technical errors, and unknown errors. Human interaction errors were noticed during researcher-patient or researcher-nurse contact moments. Technical errors were noticed during the iterative development process of the technological platform. Note that multiple errors can be reported for a single measurement day. To determine the development of errors over time, a linear regression analysis was performed between the number of errors (technical and human interaction) as the dependent variable and the number of participants as the independent variable. Furthermore, the number of patients who are non-willing to participate, the number of dropouts, and the patient experience of the IMUs data-collection approach were determined. The patients’ experience of the data-collection approach was assessed using a questionnaire regarding the feasibility of the IMU-sensors. Surveyed topics of this questionnaire were related to the easiness of use, impediment by the sensors, willingness to wear the sensors, and categories of interest related to the possibilities of sensor-based data collection. Based on the study of Mansfield et al. [5] a percentage of ≤ 5% of missing days or nights due to technical as well as human interaction errors was assumed feasible. Dropout rates will be considered as feasible when less than 10% of the eligible patients will dropout [18]. Descriptive statistics are presented by mean ± SD if data are normally distributed, otherwise by median and range (min-max).

## 3. Results

### 3.1. Participants

Thirty-six patients were found eligible for participation in the study. Eleven (30%) patients were not willing to participate in the study. Reasons for not participating were: too much cognitive load (3 iSCI/2 stroke) e.g., too much to think about and the idea that they have to wear something around the ankles, not willing to wear the sensors (1 iSCI/1 stroke), not gaining anything from it (1 iSCI), no reasons related to this study (2 stroke) or no reason given (1 iSCI). A total of 25 rehabilitation patients were included in this study, consisting of 12 iSCI patients and 13 first-ever stroke survivors. One stroke participant dropped-out during the first day, because of the additional cognitive load to participate in the study. Therefore, the data in this study comprises the data of 24 rehabilitation patients; 12 iSCI and 12 stroke patients. 

### 3.2. Measurement Days and Nights

A total of 147 days were measured with a median of 6 (5–8) days per participant. On 114 (78%) of the 147 measurement days, the sensor data were uploaded automatically to the Cloud. Only 67 (46%) of these days were automatically pre-processed and processed by the platform resulting in gait data. The remaining 80 days (147–67 days) were manually adjusted (reprocessing or reassembly) and again uploaded to the Cloud during the iterative process of the platform development. After optimization of the process, 59 (40%) days of the 80 days could be automatically processed. The remaining 21 (14%) days were considered as lost resulting in 126 (86%) days on which data was obtained with a median of 5.5 (1–8) days per participant. The average hours of measured IMU sensor data per day were 11.1 ± 2.0 h per day excluding the first and last days. On the first and last day of the measurement week, the IMUs may have been worn for a shorter period due to the study protocol and therefore not incorporated in the average hours calculation. For the bed sensor, data of 125 (96%) nights were obtained from the total of 130 measurement nights with a median of 5 (3–7) nights per participant. Table 1 provides an overview of the percentages for missing measurement data. 

### 3.3. Error Causes

A total number of 133 errors were reported consisting of 39 errors caused by human interaction, 91 errors were caused due to technical reasons and 3 unknown errors. In Figure 3, the total number of reported errors is shown for each participant individually. Over time, the number of technical errors significantly decreased, whereas no significant increase was seen in the number of human interaction errors. A significant negative correlation (R^2^ = 0.24, *p* = 0.016) with a slope of −0.16 and intercept of 5.8 was found for the technical errors. For human interaction errors, there was no significant correlation (R^2^ = 0.11, *p* = 0.11) with a positive slope of 0.07. There were 54 measurement days without any reported error, 60 days with one reported error, and 33 days with more than one reported error.

The various technical errors encountered in this study and the summation of technical, human and unknown errors (in bold) are shown in Table 2. Only a few reported errors led to lost data, most of the reported errors could be solved. The root technical error cause resulting in lost measurement days was the absence of left and/or right IMU data (n = 14). Human interaction errors leading to a loss in measurement days were caused by an internal IMU error (1x) and switching of the NUC (2x). The other 36 human interaction errors, not leading to lost measurement days, encountered; IMU turned off (2x), IMU battery empty (7x), IMU incorrect placed (13x), problems with straps (4x), internal IMU error (1x), USB-cables detached (1x), and forgotten to wear the IMUs (2x). The 21 lost measurement days arose in eleven participants (6 iSCI/5 stroke), ranging from 1 to 4 lost days per participant (5x 1, 3x 2, 2x 3, and 1x 4 lost days). 

### 3.4. User Experience

Table 3 presents the results of the feasibility questionnaire regarding the use of IMUs. No significant differences were seen between iSCI and stroke patients (tested with Fisher’s exact test). Five of the 24 participants needed assistance from the nursing staff to put on the IMUs. Putting on the IMUs was then incorporated in the dressing process. Nobody was impeded by wearing the IMUs. However, two of the participants mentioned that remarks were made by others about their sensors, but none of them experienced this as annoying. Furthermore, participants made side notes about; the cognitive load (7x) for example the fear of forgetting and paying attention to the correct placement of the IMUs, the idea that something is around the ankles (7x), and asking the nursing staff for help (1x). Suggestions for improvement consisted of refinement of the straps (7x), reduction of the flashing LEDs on the IMUs which indicate device status and operating mode (4x), and making the IMUs waterproof (1x). A total of 20 participants indicated that they were willing to wear the IMUs during their entire admission and 18 participants were interested in results about their own functioning related to activities (n = 8), their progression (n = 7), and gait parameters (n = 6).

## 4. Discussion

The current study determined the feasibility of a sensor-based technological platform within a clinical routine setting based on missing data, error cause, and user experience. Only small percentages of missing measurement days (14%) and nights (4%) were found. The main cause of lost measurement days was related to missing data of one or two IMU sensors. Most technical errors were solved during the iterative process of this study. One patient dropped out during the first day of the measurement week due to the additional cognitive load of participation. Other participants were not impeded, did not experience any discomfort, and found the sensors easy to use. The first step to determine how a technological sensor-based platform can be integrated within the existing clinical treatment regimen has been taken.

### 4.1. Feasibility

In contrast to most previous studies using sensor technology, the real clinical setting was mimicked without interference of a researcher or research assistant. Studies focusing or reporting on missing data are scarce. However, within controlled standardized settings or inpatient studies of short duration, a minimal loss of data has been reported ranging from 0 up to 24% when using various sensor technologies [5,34,37,38,39,40,41,42]. The 4% of missing measurement nights correspond with those numbers and are within the 5% boundary for feasibility. However, the 14% of missing measurement days in the current study exceeds the 5% boundary. Nevertheless, it is still in line with previous literature despite the difference of performing the study in the clinical setting, which causes the data collection to be more prone to errors. For clinical implication (e.g., the evaluation of therapies and monitoring of rehabilitation progress) it is important that sensors can reliably measure changes over time. The amount of data required to measure these changes depends on the outcome parameter. A minimum of two to three days within a week has been shown to be necessary to reliably measure step count [43,44]. However, for physical activity, data collection over 7 days is proposed [45,46]. Depending on the information of interest, a complete measurement week is not necessarily required and fewer days might already be sufficient. In the current study, all participants except one had more than two days of obtained data. Moreover, 13 of the 24 participants had no missing days at all, and 5 had only 1 missing day over a week which is sufficient for certain gait outcome parameters. Nonetheless, to gain the best insight into patients’ rehabilitation progress, it is strived to collect as much information as possible. Hence, despite the percentage of 14% missing measurement days, sufficient data have been received to be able to find the platform clinically feasible.

### 4.2. Error Causes

Missing data resulting from technical errors was 10%, which matches with previous literature [34]. Frequently described technical errors in various wearable-sensor studies reporting on missing data are battery failure, device failure, storage problems, or connectivity issues [10,38,45,47,48,49,50,51,52,53]. The technical errors encountered in our study are in accordance with these findings. However, the decrease of technical errors over time indicates that technical hard- and software errors could be solved and the platform was optimized during the iterative process. In contrast, the number of human interaction errors did not significantly change over time. Missing data from human interaction errors was 2% which is less compared to previous literature [34]. A system with human and technology interaction is prone to errors. Not wearing, losing, and incorrect use of sensors are often reported human interaction errors [38,44,45,48,49,50,51,52,54,55] and were also reported in our study. However, those errors can be reduced to a minimum with proper management. Although human interaction errors can be expected to be dependent on type of neurological disorder, there were no between-group differences found in this study. Therefore, it might be more depending on personal interest, which is inherent to the appearance that most reasons for participation in clinical research are humanitarian in nature [56]. 

The main cause of error was the absence of left and/or right IMU data. Although we considered it as a technical error, the absence of left and/or right IMU data could also be caused by a human interaction error which might not be reported. If this is the case this is more in line with the literature, which reported a majority of missing data related to human interaction errors in contrast to a minority related to technical difficulties [5,34]. However, the possible human interaction origin cannot be determined on the basis of this study design, yet is something to be aware of. 

### 4.3. User Experience

To achieve widespread acceptance among patients, a monitoring system needs to be easy-to-use, unobtrusive, and inter-operable among various computing platforms [57]. Patients were reserved in participating in the study. Almost a third of the approached patients were not willing to participate in the study, which is in accordance with 37% in the study by Mansfield et al. [5] and 20% in the study by Weenk et al. [38]. Various barriers, including cognitive load, are known for recruitment and retention of various patient populations, including iSCI and stroke patients [56]. However, 20 of the 24 participants who participated in this study indicated that they were willing to wear the IMUs during their whole clinical admission. This suggests that patients can be reluctant to participate because of the expected cognitive load, but when participating this is not an issue anymore. If the platform is integrated within the clinical treatment regimen, the additional expected cognitive load might be reduced, making people more willing to use sensor technology.

### 4.4. Limitations

To mimic the clinical setting, the researcher did not track everything during the protocol, therefore it might be that not every human interaction error was reported. If a note was made about an error, the researcher intervened and where possible corrected the problem. Therefore, the reported percentage of missing measurement days might be an underestimation of the actual clinical implementation percentage. However, when integrated, the nursing staff or patients can report any trouble or errors regarding the measurement system to the local IT department or technical support. This is already the case for most technical devices used in hospitals. In the end, all technical errors were solved during this iterative process, only human interaction errors might still remain but are part of daily life. Other limitations regarding the execution of this study in the clinical research setting comprise the lack of participatory observations during sensor-use and the use of only one sensor regarding sleep monitoring. However, the strength of this study was to let the clinical setting run its course, and therefore it was chosen to mimic the clinical setting with as little as possible research-related interruptions and additions that could raise the awareness of the participants and clinical staff of using the sensors. Clinical implementation of the sensor-based technological platform is possible and feasible, but this is only the first step of the integration of such a platform within the existing clinical treatment regimen. It has to be investigated whether the same results can be maintained over a longer measuring period. Furthermore, next steps should be made regarding the validity of continuously obtained data and feedback loops, because there is a need for automated data analysis and categorization procedures that offer reliable and ad hoc interpretation [58].

## 5. Conclusions

The sensor-based technological platform is feasible to use within the clinical rehabilitation setting for continuously monitoring gait and sleep of iSCI and stroke patients. It is possible to implement sensor technology with minimal interference of a researcher, continuous and automatic data can be obtained, and participants are willing to wear sensors around their ankles during the clinical admission. This paves the way for tracking patients’ progression and evaluation during inpatient rehabilitation.

## Figures and Tables

**Figure 1 sensors-20-02748-f001:**
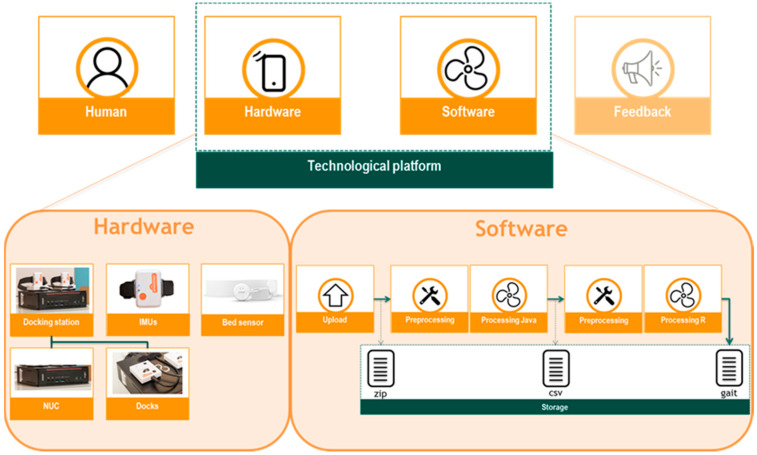
Graphical representation of the sensor-based technological platform consisting of a hardware (docking station consisting of Next Unit of Computing (NUC) and two docks, Inertial Measurement Units (IMUs), and a bed sensor) and software (upload, preprocessing, processing, and storage) component.

**Figure 2 sensors-20-02748-f002:**
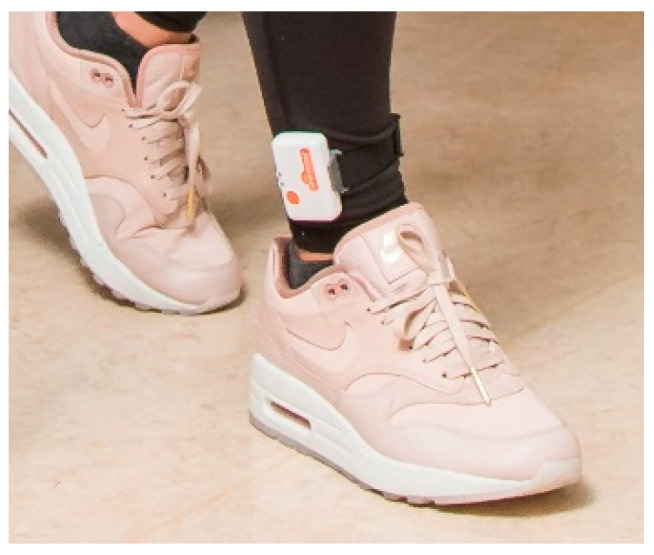
Placement of the Inertial Measurement Unit (IMU) on the lateral side of the ankle, above the malleoli.

**Figure 3 sensors-20-02748-f003:**
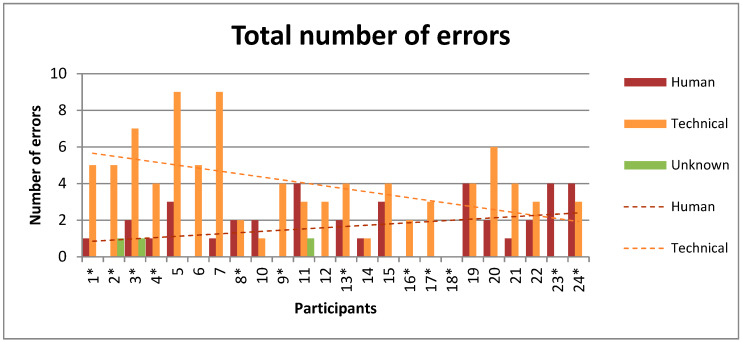
Number of technical, human interaction, and unknown errors for each participant. Participants are ordered chronologically based on inclusion date. Incomplete Spinal Cord Injury (iSCI) patients are marked with an asterisk (*), others are stroke patients. Linear regression lines based on errors over time have a significant negative correlation (R^2^ = 0.24) for technical errors and non-significant positive correlation (R^2^ = 0.11) for human interaction errors.

**Table 1 sensors-20-02748-t001:** Number and percentages of missing measurement days and nights. Measurement nights did not have any processing or reprocessing procedures and are therefore not applicable (n.a.).

	Expected	Automatically Obtained			After Reprocessing
		Upload	(pre)Processed	Lost	Lost
Days	147 (100%)	114/147 (78%)	67/147 (46%)	80/147 (54%)	21/147 (14%)
Nights	130 (100%)	125/130 (96%)	n.a.	5/130 (4%)	n.a.

**Table 2 sensors-20-02748-t002:** Encountered technical errors and total number of reported human, technical, and unknown error causes including root causes related to lost measurement days.

Target Component	Error Cause	# Reported	# Lost (Root Cause)
Human/Hardware	Data not transmitted from dock to NUC	11	1
Hardware	Loss of Wi-Fi connectivity	4	0
Hardware	NUC booting to the bios screen instead of Linux	18	0
Preprocessing	Repository not ready (R or Java)	12	0
Preprocessing	Azure batch jobs maximum reached	2	0
Processing	No left and/or right IMU data available	16	14
Processing	Data left and right IMU not matched properly by NUC	28	0
**Technical**		**91**	**15**
**Human**		**39**	**3**
**Unknown**		**3**	**3**
**Total**		**133**	**21**

NUC is Next Unit of Computing; IMU is Inertial Measurement Unit. Summation of technical, human, unknown and total errors are shown in bold.

**Table 3 sensors-20-02748-t003:** Results of the questionnaire on the feasibility regarding the use of Inertial Measurement Units (IMUs) for both patient groups. Results are displayed as total yes/no or as NRS score (median with minimum and maximum value).

	iSCI	Stroke	Total	P-value
Put IMUs on independently? (yes/no)	10/2	9/3	19/5	*P* = 0 1
Easiness of use (0–10)	9 (7–10)	8 (6–10)	8 (6–10)	*P* = 0.132
Impeded by the IMUs? (yes/no)	0/12	0/12	0/24	-
Bothered by wearing IMUs? (0–10)	0 (0–2)	0 (0–3)	0 (0–3)	*P* = 0.208
Whole clinical admission? (yes/no)	10/2	10/2	20/4	*P* = 1
Interested in activity? (yes/no)	9/3	9/3	18/6	*P* = 1

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
