# Peer review of "Feasibility of a Sensor-Based Technological Platform in Assessing Gait and Sleep of In-Hospital Stroke and Incomplete Spinal Cord Injury (iSCI) Patients"

_sensors, 2020, doi:10.3390/s20102748_

Round 1
Reviewer 1 Report
The manuscript presents a feasibility study of a sensor platform in assessing gait and sleep. It is well written and the results are promising. There are a few issues to be addressed before the paper can be accepted.
It is mentioned in the abstract that "... Clinical usage and implementation are of utmost relevance and have not been studied." Actually, D. Johansson et al., published a review article to summarize all clinical applications of wearable sensors in epilepsy, PD, and stroke in 2018. https://link.springer.com/article/10.1007/s00415-018-8786-y The authors need to do more literature work before making this statement. Also, if the authors can not defend this, the innovation of this work will be questioned.
In section 3.3 Error causes, "Over time, the number of technical errors 219 decreased, whereas no trend was seen in the number of human interaction", this claim is not supported by any data, figure, or table.
Regarding the missing data discussion, the manuscript presents a few studies to compare the percentage of missing data and claims it will be sufficient if more than 2 days of data is acquired. However, every study is different from others in terms of study design and study goals. 2 days of data cannot be used as a universal criterion. The point is to validate if the missing data has a key information component or not. Thus, this discussion cannot convince the readers, at least the reviewer, the missing data is not important.
The meanings of those items in the patients' questionnaire are not well explained. For example, "cognitive load" is not a self-explainable term.
There are also some typos and missing spaces in the paper.
The authors need to address the above issues before the paper can be accepted.
Reviewer 2 Report
The article is well written and the objective is clear. The authors followed a sound methodological approach although there is a concern related the duration of the study. One week might be barely statistically sufficient, even considering the number of subjects involved.
However, the main comment of the reviewer is the lack of a comparative framework with other literature study. Indeed, it is known that many wearable-based studies (not only in the specific application domain of this work) are still today conducted in controlled environments, but there are many others that face field trials and pilot use cases. It would be in particular interesting to compare the findings of the present study with other previous works, in terms of the metrics that are more likely to be reported, such that related to errors.
Reviewer 3 Report
The problem is well identified, clearly and objectively. The structure of the paper is well organized and the method used is appropriate for the intended study.
However, I think it was useful to explain in the introduction how movement during sleep can be an indicator of patients' physical status.
The equipment used to carry out the study seems to have been well chosen and described, however I think that the use of more than one sensor in the bed may provide more concrete data during sleep.
The user experience during the use of the platform could be complemented with participatory observation during use. The use of the questionnaire, although very useful in this type of studies, must be complemented with other forms of error assessment, users are often not aware of some of the errors they make during use.
Round 2
Reviewer 2 Report
The authors satisfied the reviewer's concerns with this revision. The paper can be accepted in the present form